# Variance Networks: When Expectation Does Not Meet Your Expectations

**Kirill Neklyudov**[*]
Samsung-HSE Laboratory, National Research
University Higher School of Economics
Samsung AI Center Moscow
k.necludov@gmail.com

**Dmitry Molchanov**[*]
Samsung-HSE Laboratory, National Research
University Higher School of Economics
Samsung AI Center Moscow
dmolch111@gmail.com

**Arsenii Ashukha**[*]
Samsung AI Center Moscow
ars.ashuha@gmail.com

**Dmitry Vetrov**
Samsung-HSE Laboratory, National Research
University Higher School of Economics
Samsung AI Center Moscow
vetrovd@yandex.ru

## Abstract

Ordinary stochastic neural networks mostly rely on the expected values of their weights to make predictions, whereas the induced noise is mostly used to capture the uncertainty, prevent overfitting and slightly boost the performance through test-time averaging. In this paper, we introduce *variance layers*, a different kind of stochastic layers. Each weight of a variance layer follows a zero-mean distribution and is only parameterized by its variance. It means that each object is represented by a zero-mean distribution in the space of the activations. We show that such layers can learn surprisingly well, can serve as an efficient exploration tool in reinforcement learning tasks and provide a decent defense against adversarial attacks. We also show that a number of conventional Bayesian neural networks naturally converge to such zero-mean posteriors. We observe that in these cases such zero-mean parameterization leads to a much better training objective than more flexible conventional parameterizations where the mean is being learned.

## 1 Introduction

Modern deep neural networks are usually trained in a stochastic setting. They use different stochastic layers (Srivastava et al. (2014); Wan et al. (2013)) and stochastic optimization techniques (Welling & Teh (2011); Kingma & Ba (2014)). Stochastic methods are used to reduce overfitting (Srivastava et al. (2014); Wang & Manning (2013); Wan et al. (2013)), estimate uncertainty (Gal & Ghahramani (2016); Malinin & Gales (2018)) and to obtain more efficient exploration for reinforcement learning (Fortunato et al. (2017); Plappert et al. (2017)) algorithms.

Bayesian deep learning provides a principled approach to training stochastic models (Kingma & Welling (2013); Rezende et al. (2014)). Several existing stochastic training procedures have been reinterpreted as special cases of particular Bayesian models, including, but not limited to different versions of dropout (Gal & Ghahramani (2016)), drop-connect (Kingma et al. (2015)), and even the stochastic gradient descent itself (Smith & Le (2018)). One way to create a stochastic neural network from an existing deterministic architecture is to replace deterministic weights $w_{ij}$ with random weights $\hat{w}_{ij} \sim q(\hat{w}_{ij} \,|\, \phi_{ij})$ (Hinton & Van Camp (1993); Blundell et al. (2015)). During training, a distribution over the weights is learned instead of a single point estimate. Ideally one would want to average the predictions over different samples of such distribution, which is known as test-time averaging, model averaging or ensembling. However, test-time averaging is impractical, so during inference the learned distribution is often discarded, and only the expected values of the weights are used instead. This heuristic is known as mean propagation or the weight scaling rule

---
[*]Equal contribution

(Srivastava et al. (2014); Goodfellow et al. (2016)), and is widely and successfully used in practice (Srivastava et al. (2014); Kingma et al. (2015); Molchanov et al. (2017)).

In our work we study the an extreme case of stochastic neural network where all the weights in one or more layers have zero means and trainable variances, e.g. $w_{ij} \sim \mathcal{N}(0, \sigma_{ij}^2)$. Although no information get stored in the expected values of the weights, these models can learn surprisingly well and achieve competitive performance. Our key results can be summarized as follows:

1. We introduce variance layers, a new kind of stochastic layers that store information only in the variances of its weights, keeping the means fixed at zero, and mapping the objects into zero-mean distributions over activations. The variance layer is a simple example when the weight scaling rule (Srivastava et al. (2014)) fails.

2. We draw the connection between neural networks with variance layers (variance networks) and conventional Bayesian deep learning models. We show that several popular Bayesian models (Kingma et al. (2015); Molchanov et al. (2017)) converge to variance networks, and demonstrate a surprising effect – a less flexible posterior approximation may lead to much better values of the variational inference objective (ELBO).

3. Finally, we demonstrate that variance networks perform surprisingly well on a number of deep learning problems. They achieve competitive classification accuracy, are more robust to adversarial attacks and provide good exploration in reinforcement learning problems.

## 2 STOCHASTIC NEURAL NETWORKS

A deep neural network is a function that outputs the predictive distribution $p(t \mid x, W)$ of targets $t$ given an object $x$ and weights $W$. Recently, stochastic deep neural networks — models that exploit some kind of random noise — have become widely popular (Srivastava et al. (2014); Kingma et al. (2015)). We consider a special case of stochastic deep neural networks where the parameters $W$ are drawn from a parametric distribution $q(W \mid \phi)$. During training the parameters $\phi$ are adjusted to the training data $(X, T)$ by minimizing the sum of the expected negative log-likelihood and an optional regularization term $R(\phi)$. In practice this objective equation 1 is minimized using one-sample mini-batch gradient estimation.

$$- \mathbb{E}_{q(W \mid \phi)} \log p(T \mid X, W) + R(\phi) \to \min_{\phi} \qquad (1)$$

This training procedure arises in many conventional techniques of training stochastic deep neural networks, such as binary dropout (Srivastava et al. (2014)), variational dropout (Kingma et al. (2015)) and drop-connect (Wan et al. (2013)). The exact predictive distribution $\mathbb{E}_{q(W \mid \phi)} p(t \mid x, W)$ for such models is usually intractable. However, it can be approximated using $K$ independent samples of the weights equation 2. This technique is known as test-time averaging. Its complexity increases linearly in $K$.

$$p(t \mid x) = \mathbb{E}_{q(W \mid \phi)} p(t \mid x, W) \simeq \frac{1}{K} \sum_{k=1}^{K} p(t \mid x, \widehat{W}_k), \quad \widehat{W}_k \sim q(W \mid \phi) \qquad (2)$$

In order to obtain a more computationally efficient estimation, it is common practice to replace the weights with their expected values equation 3. This approximation is known as the weight scaling rule (Srivastava et al. (2014)).

$$\mathbb{E}_{q(W \mid \phi)} p(t \mid x, W) \approx p(t \mid x, \mathbb{E}_q W). \qquad (3)$$

As underlined by Goodfellow et al. (2016), while being mathematically incorrect, this rule still performs very well on practice. The success of weight scaling rule implies that a lot of learned information is concentrated in the expected value of the weights.

In this paper we consider symmetric weight distributions $q(W \mid \phi) = q(-W \mid \phi)$. Such distributions cannot store any information about the training data in their means as $\mathbb{E}W = 0$. In the case of conventional layers with symmetric weight distribution, the predictive distribution $p(t \mid x, \mathbb{E}W = 0)$ does not depend on the object $x$. Thus, the weight scaling rule results in a random guess quality predictions. We would refer to such layers as the *variance layers*, and will call neural networks that at least one variance layer the *variance networks*.

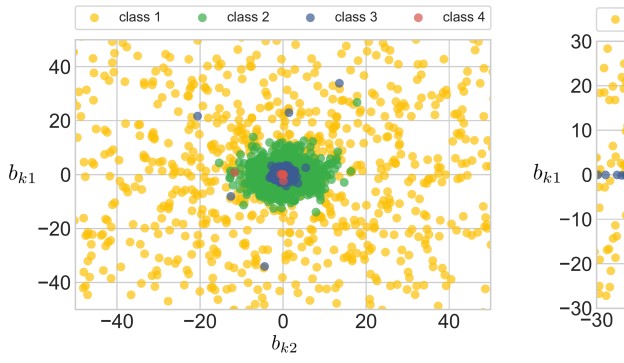 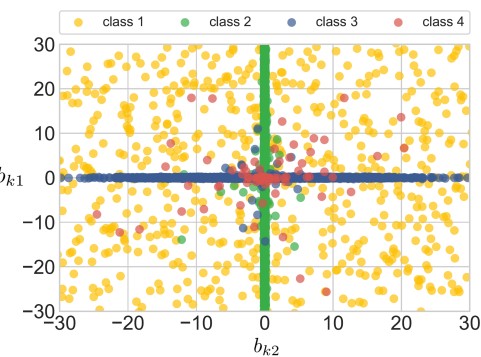

(a) Neurons encode the same information     (b) Neurons encode different information

Figure 1: The visualization of objects activation samples from a variance layer with two variance neurons. The network was learned on a toy four-class classification problem. The two plots correspond to two different random initializations. We demonstrate that a variance layer can learn two fundamentally different kinds of representations (a) two neurons repeat each other, the information about each class is encoded in variance of each neuron (b) two neurons encode an orthogonal information, both neurons are needed to identify the class of the object.

# 3 VARIANCE LAYER

In this section, we consider a single fully-connected layer[1] with $I$ input neurons and $O$ output neurons, before a non-linearity. We denote an input vector by $a_k \in R^I$ and an output vector by $b_k \in R^O$, a weight matrix by $W \in R^{I \times O}$. The output of the layer is computed as $b_k^\top = a_k^\top W$. A standard normal distributed random variable is denoted by $\epsilon \sim \mathcal{N}(0, 1)$.

Most stochastic layers mostly rely on the expected values of the weights to make predicitons. We introduce a *Gaussian variance layer* that by design cannot store any information in mean values of the weights, as opposed to conventional stochastic layers. In a Gaussian variance layer the weights follow a *zero-mean* Gaussian distribution $w_{ij} = \sigma_{ij} \cdot \epsilon_{ij} \sim \mathcal{N}(w_{ij} \,|\, 0, \sigma_{ij}^2)$, so the information can be stored only in the variances of the weights.

## 3.1 DISTRIBUTION OF ACTIVATIONS

To get the intuition of how the variance layer can output any sensible values let's take a look at the activations of this layer. A Gaussian distribution over the weights implies a Gaussian distribution over the activations (Wang & Manning (2013); Kingma et al. (2015)). This fact is used in the local reparameterization trick (Kingma et al. (2015)), and we also rely on it in our experiments.

$$b_{kj} = \underbrace{\sum_{i=1}^{I} a_{ki}\mu_{ij} + \epsilon_j \cdot \sqrt{\sum_{i=1}^{I} a_{ki}^2 \sigma_{ij}^2}.}_{\text{Conventional layer}} \qquad \underbrace{b_{kj} = \epsilon_j \cdot \sqrt{\sum_{i=1}^{I} a_{ki}^2 \sigma_{ij}^2}.}_{\text{Variance layer}} \qquad (4)$$

In Gaussian variance layer an expectation of $b_{mj}$ is exactly zero, so the first term in eq. equation 4 disappears. During training, the layer can only adjust the variances $\sum_i^I a_{ki}^2 \sigma_{ij}^2$ of the output. It means that each object is encoded by a zero-centered fully-factorized multidimensional Gaussian rather than by a single point / a non-zero-mean Gaussian. The job of the following layers is then to classify such zero-mean Gaussians with different variances. It turns out that such encodings are surprisingly robust and can be easily discriminated by the following layers.

## 3.2 Toy problem

We illustrate the intuition on a toy classification problem. Object of four classes were generated from Gaussian distributions with $\mu \in \{(3, 3), (3, 10), (10, 3), (10, 10)\}$ and identity covariance matrices. A classification network consisted of six fully-connected layers with ReLU non-linearities, where the fifth layer is a bottleneck variance layer with two output neurons. In Figure 1 we plot the activations of the variance layer that were sampled similar to equation equation 4. The exact expressions are presented in Appendix E. Different colors correspond to different classes.

We found that a variance bottleneck layer can learn two fundamentally different kinds of representations that leads to equal near perfect performance on this task. In the first case the same information is stored in two available neurons (Figure 1a). We see this effect as a kind of in-network ensembling: averaging over two samples from the same distribution results in a more robust prediction. Note that in this case the information about four classes is robustly represented by essentially only one neuron. In the second case the information stored in these two neurons is different. Each neuron can be either *activated* (i.e. have a large variance) or *deactivated* (i.e. have a low variance). Activation of the first neuron corresponds to either class 1 or class 2, and activation of the second neuron corresponds to either class 1 or class 3. This is also enough to robustly encode all four classes. Other combinations of these two cases are possible, but in principle, we see how the variances of the activations can be used to encode the same information as the means. As shown in Section 6, although this representation is rather noisy, it robustly yields a relatively high accuracy even using only one sample, and test-time averaging allows to raise it to competitive levels. We observe the same behaviour with real tasks. See Appendix D for the visualization of the embeddings, learned by a variance LeNet-5 architecture on the MNIST dataset.

One could argue that the non-linearity breaks the symmetry of the distribution. This could mean that the expected activations become non-zero and could be used for prediction. However, this is a fallacy: we argue that the correct intuition is that the model learns to distinguish activations of different variance. To prove this point, we train variance networks with antisymmetric non-linearities like (e.g. a hyperbolic tangent) without biases. That would make the mean activation of a variance layer exactly zero even after a non-linearity. See Appendix C for more details.

## 3.3 Other zero-mean symmetric distributions

Other types of variance layers may exploit different kinds of multiplicative or additive symmetric zero-mean noise distributions . These types of noise include, but not limited to:

- Gaussian variance layer: $q(w_{ij}) = \sigma_{ij} \cdot \epsilon_{ij}$, where $\epsilon_{ij} \sim \mathcal{N}(0, 1)$

- Bernoulli variance layer: $q(w_{ij}) = \theta_{ij} \cdot (2\epsilon_{ij} - 1)$, where $\epsilon_{ij} \sim Bernoulli(\frac{1}{2})$

- Uniform variance layer: $q(w_{ij}) = \theta_{ij} \cdot \epsilon_{ij}$, where $\epsilon_{ij} \sim Uniform(-1, 1)$,

In all these models the learned information is stored only in the variances of the weights. Applied to these type of models, the weight scaling rule (eq. equation 3) will result in the random guess performance, as the mean of the weights is equal to zero. Note that we cannot perform an exact local reparameterization trick for Bernoulli or Uniform noise. We can however use moment-matching techniques similar to fast dropout (Wang & Manning (2013)). Under fast dropout approximation all three cases would be equivalent to a Gaussian variance layer.

We were able to train a LeNet-5 architecture (Caffe (2014)) on the MNIST dataset with the first dense layer being a Gaussian variance layer up to 99.3 accuracy, and up to up to 98.7 accuracy with a Bernoulli or a uniform variance layer. Such gap in the performance is due to the lack of the local reparameterization trick for Bernoulli or uniform random weights. The complete results for the Gaussian variance layer are presented in Section 6.

---

[1]In this section we consider fully-connected layers for simplicity. The same applies to convolutional layers and other linear transformations. In the experiments we consider both fully-connected and convolutional variance layers.

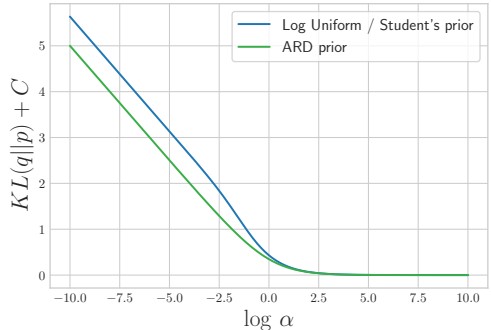
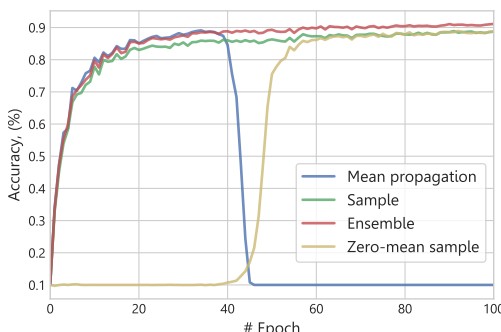

Figure 2: The KL divergence between the Gaussian dropout posterior and different priors. The KL-divergence between the Gaussian dropout posterior and the Student's t-prior is no longer a function of just $\alpha$; however, for small enough $\nu$ it is indistinguishable from the KL-divergence with the log-uniform prior.

Figure 3: CIFAR-10 test set accuracy for a VGG-like neural network with layer-wise parameterization with weights replaced by their expected values (deterministic), sampled from the variational distribution (sample), the test-time averaging (ensemble) and zero-mean approximation accuracy. Note how the information is transfered from the means to the variances between epochs 40 and 60.

## 4 RELATION TO BAYESIAN DEEP LEARNING

In this section we review several Gaussian dropout posterior models with different prior distributions over the weights. We show that the Gaussian dropout layers may converge to variance layers in practice.

### 4.1 STOCHASTIC VARIATIONAL INFERENCE

Doubly stochastic variational inference (DSVI) (Titsias & Lázaro-Gredilla (2014)) with the (local) reparameterization trick (Kingma & Welling (2013); Kingma et al. (2015)) can be considered as a special case of training with noise, described by eq. equation 1. Given a likelihood function $p(t \mid x, W)$ and a prior distribution $p(W)$, we would like to approximate the posterior distribution $p(W \mid X_{train}, T_{train}) \approx q(W \mid \phi)$ over the weights $W$. This is performed by maximization of the variational lower bound (ELBO) w.r.t. the parameters $\phi$ of the posterior approximation $q(W \mid \phi)$

$$\mathcal{L}(\phi) = \mathbb{E}_{q(W \mid \phi)} \log p(T \mid X, W) - \mathrm{KL}(q(W \mid \phi) \,\|\, p(W)) \to \max_{\phi} . \qquad (5)$$

The variational lower bound consists of two parts. One is the expected log likelihood term $\mathbb{E}_{q(W \mid \phi)} \log p(T \mid X, W)$ that reflects the predictive performance on the training set. The other is the KL divergence term $\mathrm{KL}(q(W \mid \phi) \,\|\, p(W))$ that acts as a regularizer and allows us to capture the prior knowledge $p(W)$.

### 4.2 MODELS

We consider the Gaussian dropout approximate posterior that is a fully-factorized Gaussian $w_{ij} \sim \mathcal{N}(w_{ij} \mid \mu_{ij}, \alpha \mu_{ij}^2)$ (Kingma et al. (2015)) with the Gaussian "dropout rate" $\alpha$ shared among all weights of one layer. We explore the following prior distributions:

**Symmetric log-uniform distribution** $p(w_{ij}) \propto \frac{1}{|w_{ij}|}$ is the prior used in variational dropout (Kingma et al. (2015); Molchanov et al. (2017)). The KL-term for the Gaussian dropout posterior

turns out to be a function of $\alpha$ and can be expressed as follows:

$$\mathrm{KL}(\mathcal{N}(\mu_{ij}, \alpha\mu_{ij}^2) \,\|\, \mathrm{LogU}(w_{ij})) = \tag{6}$$

$$= const - \frac{1}{2}\log\alpha\mu^2 + \frac{1}{2}\mathbb{E}_{\varepsilon\sim\mathcal{N}(0,1)}\log\mu^2\left(1 + \sqrt{\alpha}\varepsilon\right)^2 = \tag{7}$$

$$= const - \frac{1}{2}\log\alpha + \mathbb{E}_{\varepsilon\sim\mathcal{N}(0,1)}\log\left|1 + \sqrt{\alpha}\varepsilon\right| \tag{8}$$

This KL-term can be estimated using one MC sample, or accurately approximated. In our experiments, we use the approximation, proposed in Sparse Variational Dropout (Molchanov et al. (2017)).

**Student's t-distribution** with $\nu$ degrees of freedom is a proper analog of the log-uniform prior, as the log-uniform prior is a special case of the Student's t-distribution with zero degrees of freedom. As $\nu$ goes to zero, the KL-term for the Student's t-prior equation 10 approaches the KL-term for the log-uniform prior equation 7.

$$\mathrm{KL}(\mathcal{N}(\mu_{ij}, \alpha\mu_{ij}^2) \,\|\, \mathrm{St}(w_{ij}\,|\,\nu)) = \tag{9}$$

$$= const - \frac{1}{2}\log\alpha\mu^2 + \frac{\nu+1}{2}\mathbb{E}_{\varepsilon\sim\mathcal{N}(0,1)}\log\left(\nu + \mu^2(1 + \sqrt{\alpha}\varepsilon)^2\right) \tag{10}$$

As the use of the improper log-uniform prior in neural networks is questionable (Hron et al. (2017)), we argue that the Student's t-distribution with diminishing values of $\nu$ results in the same properties of the model, but leads to a proper posterior. We use one sample to estimate the expectation equation 10 in our experiments, and use $\nu = 10^{-16}$.

**Automatic Relevance Determination** prior $p(w_{ij}) = \mathcal{N}(w_{ij}\,|\,0, \lambda_{ij}^2)$ (Neal (1996); MacKay et al. (1994)) has been previously applied to linear models with DSVI (Titsias & Lázaro-Gredilla (2014)), and can be applied to Bayesian neural networks without changes. Following (Titsias & Lázaro-Gredilla (2014)), we can show that in the case of the Gaussian dropout posterior, the optimal prior variance $\lambda_{ij}^2$ would be equal to $(\alpha + 1)\mu_{ij}^2$, and the KL-term $\mathrm{KL}(q(W\,|\,\phi)\,\|\,p(W))$ would then be calculated as follows:

$$\mathrm{KL}(\mathcal{N}(\mu_{ij}, \alpha\mu_{ij}^2) \,\|\, \mathcal{N}(0, (\alpha+1)\mu_{ij}^2)) = \log\left(1 + \alpha^{-1}\right). \tag{11}$$

Note that in the case of the log-uniform and the ARD priors, the KL-divergence between a zero-centered Gaussian and the prior is constant. For the ARD prior it is trivial, as the prior distribution $p(w)$ is equal to the approximate posterior $q(w)$ and the KL is zero. For the log-uniform distribution the proof is presented in Appendix F. Note that a zero-centered posterior is optimal in terms of these KL divergences. In the next section we will show that Gaussian dropout layers with these priors can indeed converge to variance layers.

### 4.3 CONVERGENCE TO VARIANCE LAYERS

As illustrated in Figure 2, in all three cases the KL term decreases in $\alpha$ and pushes $\alpha$ to infinity. We would expect the data term to limit the learned values of $\alpha$, as otherwise the model would seem to be overregularized. Surprisingly, we find that in practice for some layers $\alpha$'s may grow to essentially infinite values (e.g. $\alpha > 10^7$). When this happens, the approximate posterior $\mathcal{N}(\mu_{ij}, \alpha\mu_{ij}^2)$ becomes indistinguishable from its zero-mean approximation $\mathcal{N}(0, \alpha\mu_{ij}^2)$, as its standard deviation $\sqrt{\alpha}|\mu_{ij}|$ becomes much larger than its mean $\mu_{ij}$. We prove that as $\alpha$ goes to infinity, the Maximum Mean Discrepancy (Gretton et al. (2012)) between the approximate posterior and its zero-mean approximation goes to zero.

**Theorem 1.** *Assume that $\alpha_t \longrightarrow +\infty$ as $t \longrightarrow +\infty$. Then the Gaussian dropout posterior $q_t(w) = \prod_{i=1}^{D}\mathcal{N}(w_i\,|\,\mu_{t,i}, \alpha_t\mu_{t,i}^2)$ becomes indistinguishable from its zero-centered approximation $q_t^0(w) = \prod_{i=1}^{D}\mathcal{N}(w_i\,|\,0, \alpha_t\mu_{t,i}^2)$ in terms of Maximum Mean Discrepancy:*

$$\mathrm{MMD}(q_t^0(w)\,\|\,q_t(w)) \leq \sqrt{\frac{2D}{\pi}}\cdot\frac{1}{\sqrt{\alpha_t}} \tag{12}$$

$$\lim_{t\to+\infty}\mathrm{MMD}(q_t^0(w)\,\|\,q_t(w)) = 0 \tag{13}$$

The proof of this fact is provided in Appendix A. It is an important result, as MMD provides an upper bound equation 38 on the change in the predictions of the ensemble. It means that we can

Table 1: Variational lower bound (ELBO), its decomposition into the data term and the KL term, and test set accuracy for different parameterizations. The test-time averaging accuracy is roughly the same for all procedures, but a clear phase transition is only achieved in layer-wise and neuron-wise parameterizations. The same result is reproduced on a VGG-like architecture on CIFAR-10; the achieved ELBO values are $-116.2$, $-233.4$ and $-1453.7$ for the layer-wise, neuron-wise and weight-wise parameterizations respectively.

| Metric | | Parameterization | | | | |
|---|---|---|---|---|---|---|
| | | zero-mean | layer | neuron | weight | additive |
| Evidence Lower Bound | $\mathcal{L}(\phi)$ | $\mathbf{-4.0}$ | $\mathbf{-17.4}$ | $\mathbf{-31.4}$ | $-602.6$ | $-227.9$ |
| Data term | $\mathbb{E}_q \log p(T \mid X, W)$ | $-4.0$ | $-15.8$ | $-17.0$ | $-33.8$ | $-31.2$ |
| Regularizer term | $\mathrm{KL}(q \parallel p)$ | $0.0$ | $1.7$ | $14.4$ | $568.8$ | $196.7$ |
| Mean propagation acc. (%) | $\hat{y} = \arg\max_t p(t \mid x, \mathbb{E}_q W)$ | $11.3$ | $11.3$ | $11.3$ | $96.6$ | $99.2$ |
| Test-time averaging acc. (%) | $\hat{y} = \arg\max_t \mathbb{E}_q p(t \mid x, W)$ | $99.4$ | $99.2$ | $99.2$ | $99.4$ | $99.2$ |

replace the learned posterior $\mathcal{N}(\mu_{ij}, \alpha\mu_{ij}^2)$ with its zero-centered approximation $\mathcal{N}(0, \alpha\mu_{ij}^2)$ without affecting the predictions of the model. In this sense we see that some layers in these models may converge to variance layers. Note that although $\alpha$ may grow to essentially infinite values, the mean and the variance of the corresponding Gaussian distribution remain finite. In practice, as $\alpha$ tends to infinity, the means $\mu_{ij}$ tend to zero, and the variances $\sigma_{ij}^2 = \alpha\mu_{ij}^2$ converge to finite values.

During the beginning of training, the Gaussian dropout rates $\alpha$ are low, and the weights can be replaced with their expectations with no accuracy degradation. After the end of training, the dropout rates are essentially infinite, and all information is stored in the variances of the weights. In these two regimes the network behave very differently. If we track the train or test accuracy during training we can clearly see a kind of "phase transition" between these two regimes. See Figure 3 for details. We observe the same results for all mentioned prior distributions. The corresponding details are presented in Appendix B.

## 5 Avoiding local optima

In this section we show how different parameterizations of the Gaussian dropout posterior influence the value of the variational lower bound and the properties of obtained solution. We consider the same objective that is used in sparse variational dropout model (Molchanov et al. (2017)), and consider the following parameterizations for the approximate posterior $q(w_{ij})$:

$$
\begin{array}{cccccc}
& \text{zero-mean} & \text{layer-wise} & \text{neuron-wise} & \text{weight-wise} & \text{additive} \\
q(w_{ij}) & \mathcal{N}(0, \sigma_{ij}^2) & \mathcal{N}(\mu_{ij}, \alpha\mu_{ij}^2) & \mathcal{N}(\mu_{ij}, \alpha_j\mu_{ij}^2) & \mathcal{N}(\mu_{ij}, \alpha_{ij}\mu_{ij}^2) & \mathcal{N}(\mu_{ij}, \sigma_{ij}^2)
\end{array} \quad (14)
$$

Note that the additive and weight-wise parameterizations are equivalent and that the layer- and the neuron-wise parameterizations are their less flexible special cases. We would expect that a more flexible approximation would result in a better value of variational lower bound. Surprisingly, in practice we observe exactly the opposite: the simpler the approximation is, the better ELBO we obtain.

The optimal value of the KL term is achieved when all $\alpha$'s are set to infinity, or, equivalently, the mean is set to zero, and the variance is nonzero. In the weight-wise and additive parameterization $\alpha$'s for some weights get stuck in low values, whereas simpler parameterizations have all $\alpha$'s converged to effectively infinite values. The KL term for such flexible parameterizations is orders of magnitude worse, resulting in a much lower ELBO. See Table 1 for further details. It means that a more flexible parameterization makes the optimization problem much more difficult. It potentially introduces a large amount of poor local optima, e.g. sparse solutions, studied in sparse variational dropout (Molchanov et al. (2017)). Although such solutions have lower ELBO, they can still be very useful in practice.

## 6 Experiments

We perform experimental evaluation of variance networks on classification and reinforcement learning problems. Although all learned information is stored only in the variances, the models perform surprisingly well on a number of benchmark problems. Also, we found that variance networks

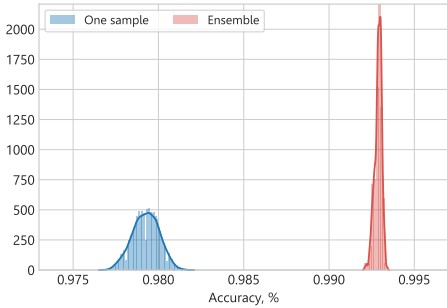 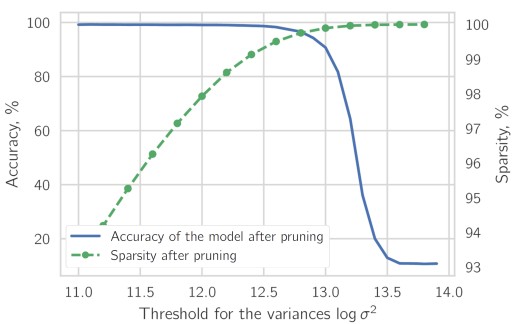

Figure 4: Histograms of test accuracy of LeNet-5 networks on MNIST dataset. The blue histogram shows an accuracy of individual weight samples. The red histogram demonstrates that test time averaging over 200 samples significantly improves accuracy.

Figure 5: Here we show that a variance layer can be pruned up to $98\%$ with almost no accuracy degradation. We use magnitude-based pruning for $\sigma$'s (replace all $\sigma_{ij}$ that are below the threshold with zeros), and report test-time-averaging accuracy.

are more resistant to adversarial attacks than conventional ensembling techniques. All stochastic models were optimized with only one noise/weight sample per step. Experiments were implemented using PyTorch (Paszke et al. (2017)). The code is available at https://github.com/da-molchanov/variance-networks.

## 6.1 CLASSIFICATION

We consider three image classification tasks, the MNIST (LeCun et al. (1998)), CIFAR-10 and CIFAR-100 (Krizhevsky & Hinton (2009)) datasets. We use the LeNet-5-Caffe architecture (Caffe (2014)) as a base model for the experiments on the MNIST dataset, and a VGG-like architecture (Zagoruyko (2015)) on CIFAR-10/100. As can be seen in Table 2, variance networks provide the same level of test accuracy as conventional binary dropout. In the variance LeNet-5 all 4 layers are variational dropout layers with layer-wise parameterization equation 14. Only the first fully-connected layer converged to a variance layer. In the variance VGG the first fully-connected layer and the last three convolutional layers are variational dropout layers with layer-wise parameterization equation 14.

Table 2: Test set classification accuracy for different methods and datasets. "Variance" stands for variational dropout model in the layer-wise parameterization equation 14. "1 samp." corresponds to the accuracy of one sample of the weights, "Det." corresponds to mean propagation, and "20 samp." corresponds to the MC estimate of the predictive distribution using 20 samples of the weights.

| Architecture | Dataset | Network | Accuracy (%) | | |
|---|---|---|---|---|---|
| | | | 1 samp. | Det. | 20 samp. |
| LeNet5 | MNIST | Dropout | 99.1 | 99.4 | 99.4 |
| | | Variance | 98.2 | 11.3 | 99.3 |
| VGG-like | CIFAR10 | Dropout | 91.0 | 93.1 | 93.4 |
| | | Variance | 91.3 | 10.0 | 93.4 |
| VGG-like | CIFAR100 | Dropout | 77.5 | 79.8 | 81.7 |
| | | Variance | 76.9 | 5.0 | 82.2 |

All these layers converged to variance layers. For the first dense layer in LeNet-5 the value of $\log \alpha$ reached 6.9, and for the VGG-like architecture $\log \alpha > 15$ for convolutional layers and $\log \alpha > 12$ for the dense layer. As shown in Figure 4, all samples from the variance network posterior robustly yields a relatively high classification accuracy. In Appendix D we show that the intuition provided for the toy problem still holds for a convolutional network on the MNIST dataset.

Similar to conventional pruning techniques (Han et al. (2015); Wen et al. (2016)), we can prune variance layer in LeNet5 by the value of weights variances. Weights with low variances have small contributions into the variance of activation and can be ignored. In Figure 5 we show sparsity and accuracy of obtained model for different threshold values. Accuracy of the model is evaluated by test time averaging over 20 random samples. Up to $98\%$ of the layer parameters can be zeroed out with no accuracy degradation.

## 6.2 REINFORCEMENT LEARNING

Recent progress in reinforcement learning shows that parameter noise may provide efficient exploration for a number of reinforcement learning algorithms (Fortunato et al. (2017); Plappert et al. (2017)). These papers utilize different types of Gaussian noise on the parameters of the model. However, increasing the level of noise while keeping expressive ability may lead to better exploration. In this section, we provide a proof-of-concept result for exploration with *variance network parameter noise* on two simple gym (Brockman et al. (2016)) environments with discrete action space: the `CartPole-v0` (Barto et al. (1983)) and the `Acrobot-v1` (Sutton (1996)). The approach we used is a policy gradient proposed by (Williams (1992); Sutton et al. (2000)).

In all experiments the policy was approximated with a three layer fully-connected neural network containing 256 neurons on each hidden layer. *Parameter noise* and *variance network* policies had the second hidden layer to be parameter noise (Fortunato et al. (2017); Plappert et al. (2017)) and variance (Section 3) layer respectively. For both methods we made a gradient update for each episode with individual samples of noise per episode. Stochastic gradient learning is performed using Adam (Kingma & Ba (2014)). Results were averaged over nine runs with different random seeds. Figure 6 shows the training curves. Using the *variance layer parameter noise* the algorithm progresses slowly but tends to reach a better final result.

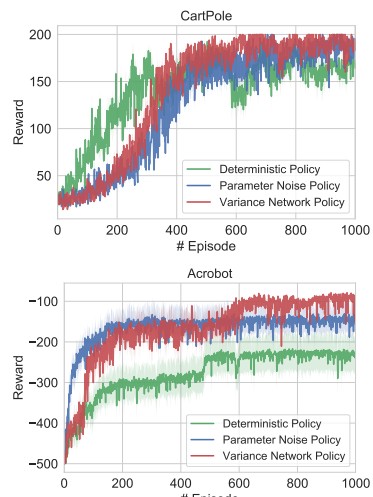

Figure 6: Evaluation mean scores for CartPole-v0 and Acrobot-v1 environments obtained by policy gradient after each episode of training for deterministic, parameter noise (Fortunato et al. (2017); Plappert et al. (2017)) and variance network policies (Section 3).

## 6.3 ADVERSARIAL EXAMPLES

Deep neural networks suffer from adversarial attacks (Goodfellow et al. (2014)) — the predictions are not robust to even slight deviations of the input images. In this experiment we study the robustness of variance networks to targeted adversarial attacks.

The experiment was performed on CIFAR-10 (Krizhevsky & Hinton (2009)) dataset on a VGG-like architecture (Zagoruyko (2015)). We build target adversarial attacks using the iterative fast sign algorithm (Goodfellow et al. (2014)) with a fixed step length $\varepsilon = 0.5$, and report the successful attack rate (Figure 7). We compare our approach to the following baselines: a dropout network with test time averaging (Srivastava et al. (2014)), and deep ensembles (Lakshminarayanan et al. (2017)) and a deterministic network. We average over 10 samples in ensemble inference techniques. Deep

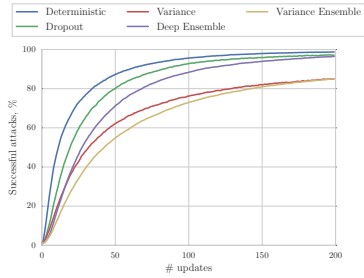

Figure 7: Results on iterative fast sign adversarial attacks for VGG-like architecture on CIFAR-10 dataset. For each iteration we report the successful attack rate. Deep ensemble of variance networks has a lower successful attacks rate.

ensembles were constructed from five separate networks. All methods were trained without adversarial training (Goodfellow et al. (2014)). Our experiments show that variance network has better resistance to adversarial attacks. We also present the results with deep ensembles of variance networks (denoted variance ensembles) and show that these two techniques can be efficiently combined to improve the robustness of the network even further.

## 7 DISCUSSION

In this paper we introduce variance networks, surprisingly stable stochastic neural networks that learn only the variances of the weights, while keeping the means fixed at zero in one or several layers.

We show that such networks can still be trained well and match the performance of conventional models. Variance networks are more stable against adversarial attacks than conventional ensembling techniques, and can lead to better exploration in reinforcement learning tasks.

The success of variance networks raises several counter-intuitive implications about the training of deep neural networks:

- DNNs not only can withstand an extreme amount of noise during training, but can actually store information using only the variances of this noise. The fact that all samples from such zero-centered posterior yield approximately the same accuracy also provides additional evidence that the landscape of the loss function is much more complicated than was considered earlier (Garipov et al. (2018)).

- A popular trick, replacing some random variables in the network with their expected values, can lead to an arbitrarily large degradation of accuracy — up to a random guess quality prediction.

- Previous works used the signal-to-noise ratio of the weights or the layer output to prune excessive units (Blundell et al. (2015); Molchanov et al. (2017); Neklyudov et al. (2017)). However, we show that in a similar model weights or even a whole layer with an exactly zero SNR (due to the zero mean output) can be crucial for prediction and can't be pruned by SNR only.

- We show that a more flexible parameterization of the approximate posterior does not necessarily yield a better value of the variational lower bound, and consequently does not necessarily approximate the posterior distribution better.

We believe that variance networks may provide new insights on how neural networks learn from data as well as give new tools for building better deep models.

### ACKNOWLEDGMENTS

We would like to thank Max Welling and Ekaterina Lobacheva for valuable discussions and feedback on the earliest version of this paper. Kirill Neklyudov and Dmitry Molchanov were supported by Samsung Research, Samsung Electronics.

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

## A  PROOF OF THEOREM 1

**Theorem 1.**  Assume that $\lim_{t\to+\infty} \alpha_t = +\infty$. Then the Gaussian dropout posterior $q_t(w) = \prod_{i=1}^{D} \mathcal{N}(w_i \mid \mu_{t,i}, \alpha_t \mu_{t,i}^2)$ becomes indistinguishable from its zero-centered approximation $q_t^0(w) = \prod_{i=1}^{D} \mathcal{N}(w_i \mid 0, \alpha_t \mu_{t,i}^2)$ in terms of Maximum Mean Discrepancy:

$$\mathrm{MMD}(q_t^0(w) \,\|\, q_t(w)) \leq \sqrt{\frac{2D}{\pi}} \cdot \frac{1}{\sqrt{\alpha_t}} \tag{15}$$

$$\lim_{t\to+\infty} \mathrm{MMD}(q_t^0(w) \,\|\, q_t(w)) = 0 \tag{16}$$

*Proof.*  By the definition of the Maximum Mean Discrepancy, we have

$$\mathrm{MMD}(q_t^0(w) \,\|\, q_t(w)) = \sup_{\substack{f\in C \\ \|f\|_\infty \leq 1}} \mathbb{E}_{q_t^0(w)} f(w) - \mathbb{E}_{q_t(w)} f(w), \tag{17}$$

where the supremum is taken over the set of continuous functions, bounded by 1. Let's reparameterize and join the expectations:

$$\sup_{\substack{f\in C \\ \|f\|_\infty \leq 1}} \mathbb{E}_{q_t^0(w)} f(w) - \mathbb{E}_{q_t(w)} f(w) = \sup_{\substack{f\in C \\ \|f\|_\infty \leq 1}} \mathbb{E}_{\varepsilon\sim\mathcal{N}(0,I_D)} \left[ f(\sqrt{\alpha_t}\mu_t \odot \varepsilon) - f(\mu_t + \sqrt{\alpha_t}\mu_t \odot \varepsilon) \right] \tag{18}$$

Since linear transformations of the argument do not change neither the norm of the function, nor its continuity, we can hide the component-wise multiplication of $\varepsilon$ by $\sqrt{\alpha_t}\mu_t$ inside the function $f(\varepsilon)$. This would not change the supremum.

$$\sup_{\substack{f\in C \\ \|f\|_\infty \leq 1}} \mathbb{E}_{\varepsilon\sim\mathcal{N}(0,I_D)} \left[ f(\sqrt{\alpha_t}\mu_t \odot \varepsilon) - f(\mu_t + \sqrt{\alpha_t}\mu_t \odot \varepsilon) \right] = \tag{19}$$

$$= \sup_{\substack{f\in C \\ \|f\|_\infty \leq 1}} \mathbb{E}_{\varepsilon\sim\mathcal{N}(0,I_D)} \left[ f(\varepsilon) - f\left( \frac{1}{\sqrt{\alpha_t}} + \varepsilon \right) \right] \tag{20}$$

There exists a rotation matrix $R$ such that $R(\frac{1}{\sqrt{\alpha_t}}, \ldots, \frac{1}{\sqrt{\alpha_t}})^\top = (\frac{\sqrt{D}}{\sqrt{\alpha_t}}, 0, \ldots, 0)^\top$. As $\varepsilon$ comes from an isotropic Gaussian $\varepsilon \sim \mathcal{N}(0, I_D)$, its rotation $R\varepsilon$ would follow the same distribution $R\varepsilon \sim \mathcal{N}(0, I_D)$. Once again, we can incorporate this rotation into the function $f$ without affecting the supremum.

$$\sup_{\substack{f\in C \\ \|f\|_\infty \leq 1}} \mathbb{E}_{\varepsilon\sim\mathcal{N}(0,I_D)} \left[ f(\varepsilon) - f\left( \frac{1}{\sqrt{\alpha_t}} + \varepsilon \right) \right] = \tag{21}$$

$$= \sup_{\substack{f\in C \\ \|f\|_\infty \leq 1}} \mathbb{E}_{\varepsilon\sim\mathcal{N}(0,I_D)} \left[ f(R^\top R\varepsilon) - f\left( R^\top R\left( \frac{1}{\sqrt{\alpha_t}} + \varepsilon \right) \right) \right] = \tag{22}$$

$$= \sup_{\substack{f\in C \\ \|f\|_\infty \leq 1}} \mathbb{E}_{\varepsilon\sim\mathcal{N}(0,I_D)} \left[ f(R\varepsilon) - f\left( R\left( \frac{1}{\sqrt{\alpha_t}} + \varepsilon \right) \right) \right] = \tag{23}$$

$$= \sup_{\substack{f\in C \\ \|f\|_\infty \leq 1}} \mathbb{E}_{\varepsilon\sim\mathcal{N}(0,I_D)} \left[ f(R\varepsilon) - f\left( \left( \frac{\sqrt{D}}{\sqrt{\alpha_t}}, 0, \ldots, 0 \right)^\top + R\varepsilon \right) \right] = \tag{24}$$

$$= \sup_{\substack{f\in C \\ \|f\|_\infty \leq 1}} \mathbb{E}_{\substack{\hat\varepsilon = R\varepsilon \\ \varepsilon\sim\mathcal{N}(0,I_D)}} \left[ f(\hat\varepsilon) - f\left( \left( \frac{\sqrt{D}}{\sqrt{\alpha_t}}, 0, \ldots, 0 \right)^\top + \hat\varepsilon \right) \right] = \tag{25}$$

$$= \sup_{\substack{f\in C \\ \|f\|_\infty \leq 1}} \mathbb{E}_{\varepsilon\sim\mathcal{N}(0,I_D)} \left[ f(\varepsilon) - f\left( \frac{\sqrt{D}}{\sqrt{\alpha_t}} + \varepsilon_1, \varepsilon_2, \ldots, \varepsilon_D \right) \right] \tag{26}$$

Let's consider the integration over $\varepsilon_1$ separately ($\phi(\varepsilon_1)$ denotes the density of the standard Gaussian distribution):

$$\sup_{\substack{f \in C \\ \|f\|_\infty \leq 1}} \mathbb{E}_{\varepsilon \sim \mathcal{N}(0, I_D)} \left[ f(\varepsilon) - f\left( \frac{\sqrt{D}}{\sqrt{\alpha_t}} + \varepsilon_1, \varepsilon_2, \dots, \varepsilon_D \right) \right] = \tag{27}$$

$$= \sup_{\substack{f \in C \\ \|f\|_\infty \leq 1}} \mathbb{E}_{\varepsilon \backslash 1 \sim \mathcal{N}(0, I_{D-1})} \int_{-\infty}^{+\infty} \left[ f(\varepsilon) - f\left( \frac{\sqrt{D}}{\sqrt{\alpha_t}} + \varepsilon_1, \varepsilon_2, \dots, \varepsilon_D \right) \right] \phi(\varepsilon_1) d\varepsilon_1 = \tag{28}$$

Next, we view $f(\varepsilon_1, \dots)$ as a function of $\varepsilon_1$ and denote its antiderivative as $F^1(\varepsilon) = \int f(\varepsilon) d\varepsilon_1$. Note that as $f$ is bounded by 1, hence $F^1$ is Lipschitz in $\varepsilon_1$ with a Lipschitz constant $L = 1$. It would allow us to bound its deviation $\left| F^1(\varepsilon) - F^1\left( \frac{\sqrt{D}}{\sqrt{\alpha_t}} + \varepsilon_1, \varepsilon_2, \dots, \varepsilon_D \right) \right| \leq \frac{\sqrt{D}}{\sqrt{\alpha_t}}$.

Let's use integration by parts:

$$\sup_{\substack{f \in C \\ \|f\|_\infty \leq 1}} \mathbb{E}_{\varepsilon \backslash 1 \sim \mathcal{N}(0, I_{D-1})} \int_{-\infty}^{+\infty} \left[ f(\varepsilon) - f\left( \frac{\sqrt{D}}{\sqrt{\alpha_t}} + \varepsilon_1, \varepsilon_2, \dots, \varepsilon_D \right) \right] \phi(\varepsilon_1) d\varepsilon_1 = \tag{29}$$

$$= \sup_{\substack{f \in C \\ \|f\|_\infty \leq 1}} \mathbb{E}_{\varepsilon \backslash 1 \sim \mathcal{N}(0, I_{D-1})} \left[ \left( F^1(\varepsilon) - F^1\left( \frac{\sqrt{D}}{\sqrt{\alpha_t}} + \varepsilon_1, \varepsilon_2, \dots, \varepsilon_D \right) \right) \phi(\varepsilon_1) \Big|_{\varepsilon_1 = -\infty}^{+\infty} - \tag{30}\right.$$

$$\left. - \int_{-\infty}^{+\infty} \left( F^1(\varepsilon) - F^1\left( \frac{\sqrt{D}}{\sqrt{\alpha_t}} + \varepsilon_1, \varepsilon_2, \dots, \varepsilon_D \right) \right) d\phi(\varepsilon_1) \right] \tag{31}$$

The first term is equal to zero, as $\left( F^1(\varepsilon) - F^1\left( \frac{\sqrt{D}}{\sqrt{\alpha_t}} + \varepsilon_1, \varepsilon_2, \dots, \varepsilon_D \right) \right)$ is bounded and $\phi(-\infty) = \phi(+\infty) = 0$. Using $d\phi(\varepsilon_1) = -\phi(\varepsilon_1)\varepsilon_1 d\varepsilon_1$, we obtain

$$\mathrm{MMD}(q_t^0(w) \,\|\, q_t(w)) = \tag{32}$$

$$= \sup_{\substack{f \in C \\ \|f\|_\infty \leq 1}} \mathbb{E}_{\varepsilon \backslash 1 \sim \mathcal{N}(0, I_{D-1})} \int_{-\infty}^{+\infty} \left( F^1(\varepsilon) - F^1\left( \frac{\sqrt{D}}{\sqrt{\alpha_t}} + \varepsilon_1, \varepsilon_2, \dots, \varepsilon_D \right) \right) \phi(\varepsilon_1)\varepsilon_1 d\varepsilon_1 \tag{33}$$

Finally, we can use the Lipschitz property of $F^1(\varepsilon)$ to bound this value:

$$\mathrm{MMD}(q_t^0(w) \,\|\, q_t(w)) \leq \tag{34}$$

$$\leq \sup_{\substack{f \in C \\ \|f\|_\infty \leq 1}} \mathbb{E}_{\varepsilon \backslash 1 \sim \mathcal{N}(0, I_{D-1})} \int_{-\infty}^{+\infty} \left| F^1(\varepsilon) - F^1\left( \frac{\sqrt{D}}{\sqrt{\alpha_t}} + \varepsilon_1, \varepsilon_2, \dots, \varepsilon_D \right) \right| \phi(\varepsilon_1)|\varepsilon_1| d\varepsilon_1 \leq \tag{35}$$

$$\leq \sup_{\substack{f \in C \\ \|f\|_\infty \leq 1}} \mathbb{E}_{\varepsilon \backslash 1 \sim \mathcal{N}(0, I_{D-1})} \int_{-\infty}^{+\infty} \frac{\sqrt{D}}{\sqrt{\alpha_t}} \phi(\varepsilon_1)|\varepsilon_1| d\varepsilon_1 = \sqrt{\frac{2D}{\pi}} \cdot \frac{1}{\sqrt{\alpha_t}} \tag{36}$$

Thus, we obtain the following bound on the MMD:

$$\mathrm{MMD}(q_t^0(w) \,\|\, q_t(w)) \leq \sqrt{\frac{2D}{\pi}} \cdot \frac{1}{\sqrt{\alpha_t}} \tag{37}$$

This bound goes to zero as $\alpha_t$ goes to infinity. $\qquad\square$

As the output of a softmax network lies in the interval $[0, 1]$, we obtain the following bound on the deviation of the prediction of the ensemble after applying the zero-mean approximation:

$$\left| \mathbb{E}_{q_t^0(w)} p(t \,|\, x, w) - \mathbb{E}_{q_t(w)} p(t \,|\, x, w) \right| \leq \sqrt{\frac{D}{2\pi}} \cdot \frac{1}{\sqrt{\alpha_t}} \quad \forall t, \forall x \tag{38}$$

## B PHASE TRANSITION PLOTS FOR DIFFERENT PRIORS

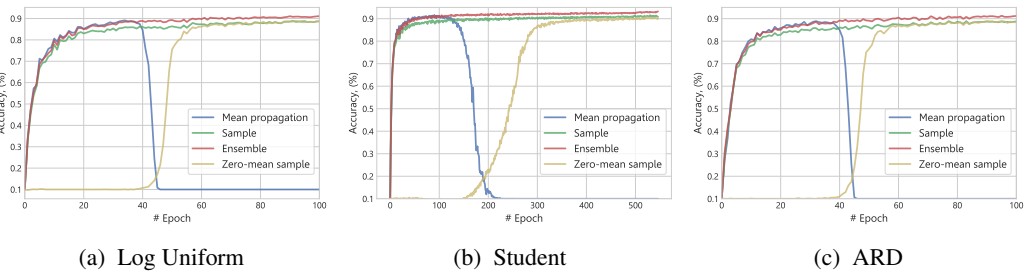

(a) Log Uniform        (b) Student        (c) ARD

Figure 8: These are the learning curves for VGG-like architectures, trained on CIFAR-10 with layer-wise parameterization and with different prior distributions. These plots show that all three priors are equivalent in practice: all three models converge to variance networks. The convergence for the Student's prior is slower, because in this case the KL-term is estimated using one-sample MC estimate. This makes the stochastic gradient w.r.t. $\log \alpha$ very noisy when $\alpha$ is large.

## C ANTI-SYMMETRIC NON-LINEARITIES

We have considered the following setting. We used a LeNet-5 network on the MNIST dataset with only tanh non-linearities and with no biases.

Table 3: One-sample (stochastic) and test-time-averaging (ensemble) accuracy of LeNet-5 tanh networks. "det" stands for conventional deterministic layers and "**var**" stands for variance layers (layers with zero-mean parameterization).

| conv1 | conv2 | dense1 | dense2 | Stochastic | Ensemble |
|-------|-------|--------|--------|------------|----------|
| det | det | det | det | 99.4 | 99.4 |
| det | **var** | det | det | 98.8 | 99.4 |
| det | det | **var** | det | 98.8 | 99.2 |
| det | **var** | **var** | det | 93.2 | 97.5 |

Note that it works well even if the second-to-last layer is a variance layer. It means that the zero-mean variance-only encodings are robustly discriminated using a linear model.

## D MNIST VARIANCE ACTIVATIONS

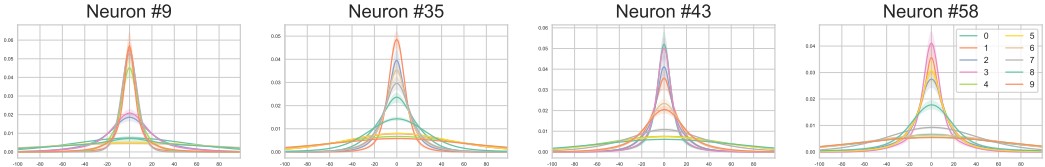

Figure 9: Average distributions of activations of a variance layer for objects from different classes for four random neurons. Each line corresponds to an average distribution, and the filled areas correspond to the standard deviations of these p.d.f.s. Each neuron essentially has several "energy levels", one for each class / a group of classes. On one "energy level" the samples have roughly the same average magnitude, and samples with different magnitudes can easily be told apart with successive layers of the neural network.

## E  LOCAL REPARAMETERIZATION TRICK FOR VARIANCE NETWORKS

Here we provide the expressions for the forward pass through a fully-connected and a convolutional variance layer with different parameterizations.

Fully-connected layer, $q(w_{ij}) = \mathcal{N}(\mu_{ij}, \alpha_{ij}\mu_{ij}^2)$:

$$b_j = \sum_{i=1}^{D_{in}} \mu_{ij}a_i + \varepsilon_j \sqrt{\sum_{i=1}^{D_{in}} \alpha_{ij}\mu_{ij}^2 a_i^2} \tag{39}$$

Fully-connected layers, $q(w_{ij}) = \mathcal{N}(0, \sigma_{ij}^2)$:

$$b_j = \varepsilon_j \sqrt{\sum_{i=1}^{D_{in}} a_i^2 \sigma_{ij}^2} \tag{40}$$

For fully-connected layers $\varepsilon_j \sim \mathcal{N}(0, 1)$ and all variables mentioned above are scalars.

Convolutional layer, $q(w_{ijhw}) = \mathcal{N}(\mu_{ijhw}, \alpha_{ijhw}\mu_{ijhw}^2)$:

$$b_j = A_i \star \mu_i + \varepsilon_j \odot \sqrt{A_i^2 \star (\alpha_i \odot \mu_i^2)} \tag{41}$$

Convolutional layers, $q(w_{ijhw}) = \mathcal{N}(0, \sigma_{ijhw}^2)$:

$$b_j = \varepsilon_j \odot \sqrt{A_i^2 \star \sigma_i^2} \tag{42}$$

In the last two equations $\odot$ denotes the component-wise multiplication, $\star$ denotes the convolution operation, and the square and square root operations are component-wise. $\varepsilon_{jhw} \sim \mathcal{N}(0, 1)$. All variables $b_j, \mu_i, A_i, \sigma_i$ are 3D tensors. For all layers $\varepsilon$ is sampled independently for each object in a mini-batch. The optimization is performed w.r.t. $\mu, \log\alpha$ or w.r.t. $\log\sigma$, depending on the parameterization.

## F  KL DIVERGENCE FOR ZERO-CENTERED PARAMETERIZATION

We show below that the KL divergence $D_{\mathrm{KL}}(\mathcal{N}(0, \sigma^2) \,\|\, \mathrm{LogU})$ is constant w.r.t. $\sigma$.

$$D_{\mathrm{KL}}(\mathcal{N}(0, \sigma^2) \,\|\, \mathrm{LogU}) \propto \tag{43}$$

$$\propto -\frac{1}{2}\log 2\pi e\sigma^2 - \mathbb{E}_{w\sim\mathcal{N}(0,\sigma^2)} \log \frac{1}{|x|} = \tag{44}$$

$$= -\frac{1}{2}\log 2\pi e\sigma^2 + \mathbb{E}_{\varepsilon\sim\mathcal{N}(0,1)} \log |\sigma\varepsilon| = \tag{45}$$

$$= -\frac{1}{2}\log 2\pi e + \mathbb{E}_{\varepsilon\sim\mathcal{N}(0,1)} \log |\varepsilon| \tag{46}$$

