# OpenReview forum: "Variance Networks: When Expectation Does Not Meet Your Expectations"
_ICLR.cc/2019/Conference_

### Official Review · AnonReviewer3 · 2018-11-01
**Stochastic neural networks with zero mean posterior on weights**

**Rating:** 6
**Confidence:** 4

**Review:**

This paper introduced a new stochastic layer termed variance layer for Bayesian deep learning, where the posterior on weight is a zero-mean symmetric distribution (e.g., Gaussian, Bernoulli, Uniform). The paper showed that under 3 different prior distributions, the Gaussian Dropout layer can converge to variance layer. Experiments verified that it can achieve similar accuracies as conventional binary dropout in image classification and reinforcement learning tasks, is more robust to adversarial attacks, and can be used to sparsify deep models.

Pros:
(1)	Proposed a new type of stochastic layer (variance layer)
(2)	Competitive performance on a variety of tasks: image classification, robustness to adversarial attacks, reinforcement learning, model compression
(3)	Theoretically grounded algorithm

Cons:
(1)	My main concern is verification. Most of the comparisons are between variance layer (zero-mean) and conventional binary dropout, while the main argument of the paper is that it’s better to set Gaussian posterior’s mean to zero. So in all the experiments the paper should compare zero-mean variance layer against variational dropout (neuron-wise Eq. 14) and sparse variational dropout (additive Eq. 14), where the mean isn’t zero.
(2)	The paper applies variance layers to some specific layers. Are there any guidelines to select which layers should be variance layers?

Some minor issues:
(1)	Page 4, equations of Gaussian/Bernoulli/Uniform variance layer, they should be w_ij=…, instead of q(w_ij)= …
(2)	What’s the prior distribution used in the experiment of Table 1?

---

> ### Author Response · Authors · 2018-11-13
> **Response to AnonReviewer3**
>
> Thank you for your review and your questions!
>
> > (1) My main concern is verification. Most of the comparisons are between variance layer (zero-mean) and conventional binary dropout, while the main argument of the paper is that it’s better to set Gaussian posterior’s mean to zero. So in all the experiments the paper should compare zero-mean variance layer against variational dropout (neuron-wise Eq. 14) and sparse variational dropout (additive Eq. 14), where the mean isn’t zero.
>
> Usually a fully-factorized Gaussian posterior achieves the same performance as the binary dropout posterior (e.g. shown in [1,2]), which we also observed in our experiments.
>
> > (2) The paper applies variance layers to some specific layers. Are there any guidelines to select which layers should be variance layers?
>
> Neural networks are usually not very stable to high amounts of noise in the first layers. Also, we have observed that it is hard to train a variance network with the last layer (right before softmax) being variance layer. Therefore a simple rule of thumb is to set the first layers to be conventional deterministic layers, then add several variance layers, and then add the last deterministic layer to obtain the logits.
>
> > (2)    What’s the prior distribution used in the experiment of Table 1?
>
> We have used the log-uniform prior in this experiment. The result for the ARD prior is the same.
>
>
> [1] Gal, Yarin, and Zoubin Ghahramani. "Dropout as a Bayesian approximation: Representing model uncertainty in deep learning." ICML 2016.
> [2] Louizos, Christos, and Max Welling. "Multiplicative normalizing flows for variational bayesian neural networks." ICML 2017

---

### Official Review · AnonReviewer2 · 2018-11-02
**variance network uses a variational distribution with zero mean, but it achieves nice performances.**

**Rating:** 6
**Confidence:** 4

**Review:**

This paper studies variance neural networks, which approximate the posterior of Bayesian neural networks with zero-mean Gaussian distributions. The inference results are surprisingly well though there is no information in the mean of the posterior. It further shows that the several variational dropout methods are closed related to the proposed method. The experiment indicates that the ELBO can actually better optimized with this restricted form of variational distribution.

The paper is clearly written and easy to follow. The technique in the paper is solid.

However, the authors might need to clarify a few questions below.


Q1:  if every transformation is antisymmetric non-linear, then it seems that the expected distribution of $t$ in (2) is zero. Is this true or not? In another word, class information has to be read out from the encoding of instances in Fig 1. It seems antisymmetric operators cannot do so, as it will only get symmetric distributions from symmetric distributions.

Q2: it is not straightforward to see why KL term needs to go zero. In my understanding, the posterior aims to fit two objectives: maximizing data likelihood and minimizing KL term. When the signal from the data is strong (e.g. large amount of data), the first objective becomes more important. Then q does not really try to make KL zero, and alpha has no reason to go infinity. Can you explain more?

Q3: Is the claimed benefit from the optimization procedure or the special structure of the variance layer? Is it possible to test the hypothesis by 1) initializing a q distribution with learnable mean by the solution of variance neural network and then 2) optimizing q? Then the optimization procedure should continue to increase ELBO. Then compare the learned q against the variance neural network. If the learned q is better than the variance network -- it means the network structure is better for optimization, but the structure itself might not be so special. If the learned q is worse than the variance network, then the structure is interesting.


A few detailed comments:

1. logU used without definition.
2. if the paper has a few sentence explaining "Gaussian dropout approximate posterior", section 4 will be smoother to read.

---

> ### Author Response · Authors · 2018-11-13
> **Response to AnonReviewer2**
>
> Thank you for your review and your questions!
>
> > Q1:  if every transformation is antisymmetric non-linear, then it seems that the expected distribution of $t$ in (2) is zero. Is this true or not? In another word, class information has to be read out from the encoding of instances in Fig 1. It seems antisymmetric operators cannot do so, as it will only get symmetric distributions from symmetric distributions.
>
> If we have antisymmetric non-linearities, the expected value of each neuron of each layer is indeed zero. This would fail at the regression task, as the expected output of the network would be always zero. However, in multiclass classification, we use softmax to obtain predictions, so the posterior predictive distribution (the expected softmax) is non-trivial and allows to obtain reasonable predictions.
>
> > Q2: it is not straightforward to see why KL term needs to go zero. In my understanding, the posterior aims to fit two objectives: maximizing data likelihood and minimizing KL term. When the signal from the data is strong (e.g. large amount of data), the first objective becomes more important. Then q does not really try to make KL zero, and alpha has no reason to go infinity. Can you explain more?
>
> Unfortunately, in VI for Bayesian neural networks, the number of parameters is much larger than the amount of data, and the data-term in the ELBO gets overwhelmed by the KL-term. Most papers on VI in BNNs use some kind of tricks to avoid that: some downscale the KL-term (e.g. [1,3]), others restrict the variance of the approximate posterior (e.g. [1,2,4]) or underfit the ELBO in other ways. We do not use such tricks in this paper. This is one reason for alpha to go to infinity. Usually, it is not possible to set the KL to zero and retain good predictive performance with conventional priors. However, for the log-uniform and the ARD priors, the Argmin(KL(q(w)||p(w))) is a broad family of distributions, the zero-mean fully-factorized Gaussians. As we show, such family is enough to achieve a good predictive performance, so the overall objective is better: the KL is set to zero and the data-term is similar to the data-term of models with the full FFG posterior.
>
> > Q3: Is the claimed benefit from the optimization procedure or the special structure of the variance layer? Is it possible to test the hypothesis by 1) initializing a q distribution with learnable mean by the solution of variance neural network and then 2) optimizing q? Then the optimization procedure should continue to increase ELBO. Then compare the learned q against the variance neural network. If the learned q is better than the variance network -- it means the network structure is better for optimization, but the structure itself might not be so special. If the learned q is worse than the variance network, then the structure is interesting.
>
> We did try to do it. The ELBO does not increase, and the network does not change: it is equivalent to fine-tuning the variances of the variance network. The variance network is a stable local optimum: if the data-term is already good enough, the KL term would prevent the means from increasing (when the mean mu is orders of magnitude smaller than the standard deviation sigma, the KL-term behaves like log|mu+eps| for a very small eps), and the data-term would not favor increasing mu in any way.
>
> [1] Kingma, Diederik P., Tim Salimans, and Max Welling. "Variational dropout and the local reparameterization trick." NIPS 2015.
> [2] Louizos, Christos, and Max Welling. "Multiplicative normalizing flows for variational bayesian neural networks." ICML 2017
> [3] Ullrich, Karen, Edward Meeds, and Max Welling. "Soft weight-sharing for neural network compression." ICLR 2017
> [4] Blundell, Charles, et al. "Weight uncertainty in neural networks." ICML 2015

---

### Official Review · AnonReviewer1 · 2018-11-02
**An interesting paper, but a few questions needed to be answered**

**Rating:** 6
**Confidence:** 3

**Review:**

This paper investigates the effects of mean of variational posterior and proposes variance layer, which only uses variance to store information.

Overally, this paper analyzes an important but not well explored topic of variational dropout methods—the mean propagation at test time, and discusses the effect of weight variance in building a variational posterior for Bayesian neural networks. This findings are interesting and I appreciate the analysis.

However, I think the claim for benefits of variance layer is not well supported. Variance layer requires test-time averaging in test time to achieve competitive accuracy, while the additive case in Eq. (14) using mean propagation achieves similar performance (e.g., the results in Table 1). The results in Sec 6 lack comparison to other Bayesian methods (e.g., the additive case in Eq. (14)).

Besides, there exists several problems which needs to be addressed.

Sec 5.
Sec 5 is a little hard to follow. Which prior is chosen to produce the results in Table 1? KL(q||p)=0 for the zero-mean case corresponds to the fact that the variational posterior equals the prior, which implies the ARD prior if I did not misunderstand. In this case, the ground truth posterior p(w|D) for different methods is different and corresponding ELBO for them are incomparable.

Sec 6.
The setting in Table 2 is also unclear. As ``Variance’’ stands for variational dropout, what does ``Dropout’’ means? The original Bernoulli dropout? Besides, I’m wondering why directly variance layer (i.e., zero-mean case in Eq. (14)) is not implemented in this case.

---

> ### Author Response · Authors · 2018-11-13
> **Response to AnonReviewer1**
>
> Thank you for your review and your questions!
>
> > I think the claim for benefits of variance layer is not well supported. Variance layer requires test-time averaging in test time to achieve competitive accuracy, while the additive case in Eq. (14) using mean propagation achieves similar performance (e.g., the results in Table 1).
>
> Most techniques for training stochastic neural networks like dropout, variational inference or MCMC require test-time averaging for good uncertainty estimation. If the inference time is crucial, one may use distillation techniques to mimic the predictive distribution of the variance network with a fast deterministic DNN. If one is only interested in the accuracy, the variance networks are probably not the best way to go.
>
> > The results in Sec 6 lack comparison to other Bayesian methods (e.g., the additive case in Eq. (14)).
>
> Usually a fully-factorized Gaussian posterior achieves the same performance as the binary dropout posterior (e.g. shown in [1, 2]), which we also observed in our experiments.
>
> > Which prior is chosen to produce the results in Table 1? KL(q||p)=0 for the zero-mean case corresponds to the fact that the variational posterior equals the prior, which implies the ARD prior if I did not misunderstand. In this case, the ground truth posterior p(w|D) for different methods is different and corresponding ELBO for them are incomparable.
>
> We have used the log-uniform prior in Table 1, however, the results for the ARD prior are the same. The result of this experiment can be discussed even without the Bayesian interpretation. Here we have 5 models with exactly the same objective function. Two of the models (weight-wise and additive) are equivalent and contain other models (neuron-wise, layer-wise, zero-mean) as special cases. We would expect more "rich" models to achieve a better value of the training objective. Surprisingly, in practice, we observe exactly the opposite.
>
> > The setting in Table 2 is also unclear. As ``Variance’’ stands for variational dropout, what does ``Dropout’’ means? The original Bernoulli dropout?
>
> Yes, we compare to plain binary (Bernoulli) dropout. ''Variance'' stands for a variance network that is trained using variational dropout (we explicitly switch to the zero-mean parameterization during test time to obtain a variance network).
>
> > Besides, I’m wondering why directly variance layer (i.e., zero-mean case in Eq. (14)) is not implemented in this case.
>
> It is hard to train variance layers from scratch, whereas the training of variational dropout in layer-wise multiplicative parameterization is stable (see Appendix B). During test-time, we explicitly use the zero-mean parameterization to ensure that we obtain a true variance network.
>
> [1] Gal, Yarin, and Zoubin Ghahramani. "Dropout as a Bayesian approximation: Representing model uncertainty in deep learning." ICML 2016.
> [2] Louizos, Christos, and Max Welling. "Multiplicative normalizing flows for variational bayesian neural networks." ICML 2017

---

### Meta-Review · Area_Chair1 · 2018-12-16
**Interesting and counter-intuitive result**

**Confidence:** 5
**Recommendation:** Accept (Poster)

**Metareview:**

The authors describe a very counterintuitive type of layer: one with mean zero Gaussian weights. They show that various Bayesian deep learning algorithms tend to converge to layers of this variety. This work represents a step forward in our understanding of bayesian deep learning methods and potentially may shine light on how to improve those methods.